# Activity Intensity and All-Cause Mortality Following Fall Injury Among Older Adults: Results from a 12-Year National Survey

**DOI:** 10.3390/healthcare13192530

**Published:** 2025-10-07

**Authors:** Oluwaseun Adeyemi, Tracy Chippendale, Olugbenga Ogedegbe, Dowin Boatright, Joshua Chodosh

**Affiliations:** 1Department of Emergency Medicine, New York University Grossman School of Medicine, New York, NY 10016, USA; dowin.boatright@nyulangone.org; 2Department of Occupational Therapy, Steinhardt School of Culture, Education, and Human Development, New York University, New York, NY 10003, USA; tracy.chippendale@nyu.edu; 3Institute for Excellence in Health Equity, New York University Grossman School of Medicine, New York, NY 10016, USA; olugbenga.ogedegbe@nyulangone.org; 4Department of Population Health, New York University Grossman School of Medicine, New York, NY 10016, USA; joshua.chodosh@nyulangone.org; 5Department of Medicine, New York University Grossman School of Medicine, New York, NY 10016, USA; 6Veterans Affairs New York Harbor Healthcare System, New York, NY 10010, USA

**Keywords:** activity intensity, older adults, fall, metabolic equivalent of task, mortality

## Abstract

Background: Fall injury is a sentinel event for mortality among older adults, and activity intensity may play a role in mitigating this outcome. This study assessed the relationship between activity intensity and all-cause mortality following fall injury among community-dwelling U.S. older adults. Methods: For this retrospective cohort study, we pooled 12 years of data from the National Health Interview Survey and identified older adults (aged 65 years and older) who sustained fall injuries (N = 2454). The outcome variable was time to death following a fall injury. We defined activity intensity as a binary variable, none-to-low and normal-to-high, using the American Heart Association’s weekly 500 Metabolic Equivalent of Task (MET) as a cutoff. We controlled for sociodemographic, healthcare access, and health characteristics; performed survey-weighted Cox proportional hazard regression analysis; and reported the adjusted mortality risks (plus 95% confidence interval (CI)). Results: The survey comprised 2454 older adults with fall injuries, representing 863,845 US older adults. The population was predominantly female (68%), non-Hispanic White (85%), and divorced/separated (54%). During the follow-up period, 45% of the study population died. Approximately 81% of the study population had low activity levels. However, between 2006 and 2017, the proportion of the study population with low physical activity decreased from 90% to 67%. After adjusting for sociodemographic, healthcare access, and health characteristics, none-to-low activity intensity was associated with 50% increased mortality risk (aHR: 1.50; 95% CI: 1.20–1.87). Conclusions: Promoting higher physical activity levels may significantly reduce the all-cause mortality risk following fall injury among older adults.

## 1. Introduction

In the United States (US), falls are a leading cause of injury-related deaths among adults aged 65 years and older [1,2]. Each year, approximately one in four older adults experiences a fall, with nearly 20 percent of these incidents leading to severe injuries such as fractures, traumatic brain injuries, or other life-threatening complications [3,4]. Fall-related injuries impose a significant public health burden, contributing to over 32,000 deaths annually, along with an estimated three million emergency department visits and approximately 800,000 hospitalizations [3]. Recent data indicate that mortality following fall injuries is increasing, with fatal fall injury rates rising from 29 to 64 per 100,000 between 1999 and 2020 [5].

Physical activity may play a critical role in reducing the risk of falls among older adults by enhancing multiple aspects of physical health that directly contribute to stability and mobility [6]. Regular engagement in moderate-to-vigorous activities, such as brisk walking, strength training, or balance exercises, has been shown to improve muscle strength, flexibility, balance, and coordination [7,8,9,10]. Exercise programs that incorporate multiple types of exercise, including balance and functional exercises, or balance, functional exercise, and resistance training, have been shown to reduce the rate of falls [11,12]. Additionally, physical activity that incorporates impact and progressive resistive exercises can positively impact bone density, which lessens the severity of injuries when falls do occur [13,14,15,16]. Together, these mechanisms make physical activity a vital component in fall prevention efforts among older adults.

Engaging in regular physical activity at various intensities has been shown to improve overall health and resilience, potentially reducing the severity of injuries sustained in a fall [17]. However, the impact of different intensities—light, moderate, or vigorous—on the likelihood of mortality following fall injury remains unclear. Light activities, such as walking or gentle stretching, may help maintain functional mobility but may not provide enough conditioning to significantly reduce falls, fractures, or mortality [11]. In contrast, moderate-to-vigorous activities, such as resistance training or aerobic exercises, can improve muscle strength, bone density, and cardiovascular health, which may help mitigate the effects of a fall and lower the mortality risk [18,19]. Yet, for some older adults, engaging in high-intensity activity could inadvertently increase fall risk or worsen outcomes due to overexertion, fatigue, or instability during more demanding movements [11,20]. Thus, further research is needed to determine how different levels of activity intensity influence the likelihood of fatal outcomes from falls and to identify safe activity guidelines tailored to the needs and capabilities of older adults.

This study aims to address this gap by examining the association between activity intensity and all-cause mortality following fall injury among U.S. older adults. We hypothesize that none-to-low-intensity physical activity is associated with a higher risk of all-cause mortality following fall injury. Given the increasing rate of falls in this population, identifying the association between activity intensity and all-cause mortality following fall injury can offer insights into areas for individual and community-level interventions, as well as guidance on policies for caring for older adults.

## 2. Methods

### 2.1. Study Population

Using the Integrated Public Use Microdata Series (IPUMS) of the National Health Interview Survey (NHIS), we combined 12 years of data, from 2006 to 2017. IPUMS is a publicly accessible resource that harmonizes NHIS data, facilitating the combination of data across years [21]. The NHIS is one of the oldest national surveys in the U.S., sampling over 35,000 households and 87,500 non-institutionalized individuals each year [22]. Participants are randomly selected from across all 50 states and the District of Columbia, utilizing a multistage sampling strategy [22]. On average, the annual household response rate is 70%, with oversampling of Black, Asian, and Hispanic populations [23].

### 2.2. Inclusion and Exclusion Criteria

The total sample population between 2006 and 2017 was 1,109,807 individuals (Figure 1). We limited the analysis to older adults aged 65 years and older (n = 146,448). We further restricted the sample to older adults who reported a fall injury within three months of the interview (n = 2274). Participants who did not experience a fall during this period were excluded [24]. Of note, injury data in IPUMS NHIS, including fall-related injuries, are structured differently from standard person-level variables. Specifically, injuries are recorded in a separate injury-episode-level file, allowing for multiple injury events per respondent. We extracted the fall injury data from this file, converted it from long to wide format to identify individuals with one or more fall episodes, and then linked the injury data to the main person-level dataset using unique person identifiers. Additionally, only individuals eligible for mortality follow-up were included (n = 2454). Those eligible for mortality follow-up represented NHIS respondents who provided sufficient identifying information for linkage with the National Center for Health Statistics to determine their mortality status [25]. IPUMS reported mortality status as assumed dead or assumed alive. The final analytic sample, therefore, consisted of 2454 individuals, with 1059 assumed to be dead and 1395 respondents assumed to be alive after 12 years of follow-up.

### 2.3. Outcome Variable

The outcome variable for this study was time to death among older adults. Death was defined as all-cause mortality occurring after a fall injury. The time component was determined as the difference between the year of mortality and the baseline year of observation. Participants who were alive at the end of the study period were right-censored.

### 2.4. Predictor Variable

The primary predictor variable was activity intensity, categorized as none-to-low and normal-to-high. Activity intensity was self-reported based on responses to the following survey items: How often do you do (1) moderate-intensity and (2) vigorous-intensity leisure-time physical activities? Responses were recorded as duration in minutes and the frequency per week. Those who reported never engaging in such physical activities were assigned a value of zero (0). Consistent with earlier studies [26,27], we calculated the metabolic equivalent of tasks (METs) using 4 and 8 as the weighted factors for moderate and vigorous physical activities, respectively. Hence, the weekly MET in minutes is calculated as follows: METs = 4 × daily duration of moderate exercise x weekly frequency of moderate exercise + 8 × daily duration of vigorous exercise x weekly frequency of vigorous exercise. Individuals with <500 weekly MET-minutes were classified as none-to-low, while those with ≥500 MET-minutes were classified as normal-to-high [28,29]. We opted to define activity intensity as a binary variable to minimize variability due to recall bias [30]. However, we performed a sensitivity analysis reclassifying activity intensity as a three-level categorical variable—none, low, and normal-to-high (discussed below). For all the regression models, the “normal-to-high” category served as the reference group.

### 2.5. Potential Confounders

Several sociodemographic, health, and healthcare access factors were included as potential confounders. Sociodemographic confounders included age (categorized as 65–74, 75–84, and 85 years and higher), sex (male or female), race/ethnicity (non-Hispanic White, non-Hispanic Black, Hispanic, and other races), nativity (born in the U.S. or not), educational attainment (high school or less, some college/associate degree, bachelor’s degree, and postgraduate education), marital status (single, married, divorced, or widowed), and poverty level (poor or not poor). Health-related confounders included smoking status (never smoked, current smoker, and former smoker), activity limitations (binary variable—yes or no, defined as having any condition or health problems causing limitations in any way), chronic disease index (computed as sum of binary indicators (reported = 1, not reported = 0) of hypertension, diabetes, chronic obstructive pulmonary disease, asthma, angina, arthritis, cancer, coronary heart disease, liver disease, and peptic ulcer disease and recoded as none, 1–2, 3–4, and 5+ chronic conditions), history of repeat fall injury (yes or no), and self-rated health (defined as a three-level categorical variable—poor, fair, and good-to-excellent). Healthcare access factors included availability of care, accessibility of care, affordability of care, and health coverage status, each categorized as yes or no.

### 2.6. Handling of Missing Data

Missing data were present across several predictor and confounding variables. The extent of missingness ranged from 0.1% to 13.3%. We conducted Little’s Covariate-Dependent Missingness test to determine if missingness was completely at random, yielding a *p*-value of 1.0, suggesting randomness [31]. To minimize bias, we performed multiple imputations using the Multiple Imputation by Chained Equations (MICE) method, generating 100 predicted values from 100 imputed datasets after 100 iterations to ensure convergence [32]. We determined the final value of each missing value by averaging the predicted values, consistent with earlier research on multiple imputations [33,34].

### 2.7. Data Analysis

Descriptive statistics were generated for the total sample and stratified by activity intensity and mortality status. Differences across activity intensity and mortality groups were assessed using the chi-square test and the log-rank test, respectively. We reported the yearly trend in the activity intensity among male and female older adults. A survey-weighted Cox Proportional Hazard Regression was used to estimate unadjusted and adjusted hazard ratios (aHR) for mortality risk, with a 95% confidence interval (95% CI), while adjusting for the NHIS’s design strata and clustering. We generated the Kaplan–Meier survival curve and the Nelson–Aalen cumulative hazard estimate curve across those with normal-to-high and none-to-low activity intensity. Furthermore, we computed the mortality incidence rates, incidence rate ratios (IRR), and the attributable deaths due to activity intensity (the fraction of exposure) in these two groups. Lastly, we performed a sensitivity analysis to observe any measurable differences in mortality risks among those classified as having no intense activity and low-intense activity. We reclassified activity intensity as a three-level categorical variable—none, low, and normal-to-high—and computed the survey-weighted adjusted hazard regression analysis. Since we pooled 12 years of data, the survey weights were calculated by dividing the sample weight variable by the number of pooled years, as recommended by the NHIS [35,36]. Statistical analyses were performed using STATA version 16 and SAS version 9.4 [37,38].

### 2.8. Ethical Concerns

This study utilized publicly available de-identified data [39]. Based on guidance from the New York University Institutional Review Board, secondary data analysis of a publicly available dataset does not require IRB approval [40].

## 3. Results

In this study, the mean age of participants was 76.6 years (95% CI: 76.3, 76.9), with the majority of participants aged 65–74 years (41.1%) (Table 1). Most participants were female (68.4%) and non-Hispanic White (85.2%). A large portion of participants (90.8%) were born in the United States, and more than half (52.1%) had an education level of high school or less. Also, 41.0% were married, 90.9% lived below the poverty line, and 66% had never smoked. Five percent reported having no accessible healthcare, while 65.2% reported having activity limitations. Additionally, 66.1% had at least one chronic disease, 5.6% had repeated fall injuries, and 40% rated their health as either poor (15.7%) or fair (24.3%). Compared with participants reporting normal-to-high activity intensity, those with none-to-low activity intensity were more likely to be older, female, have a high school education or less, be widowed, report no comorbid conditions, report activity limitations, and to have died during follow-up. Over the 12-year follow-up period, 45.3% of the sample population had died. Between 2006 and 2017, the proportion of older adults with none-to-low activity intensity gradually declined from 89.6% in 2006 to 67.2% in 2016, followed by a slight increase to 71.9% in 2017 (Figure 2A). A similar pattern of decline in none-to-low activity intensity was observed across both male and female respondents, with the reduction being more pronounced among males (Figure 2B).

The mean age at death was 79.5 years (95% CI: 79.0, 79.9) while the mean age among those who were alive was 74.2 years (95% CI: 73.8, 74.6; *p* < 0.001; Table 2). Age distribution differed significantly, with 23.4% of the deceased group being aged 65–74 years, compared to 55.9% in the alive group (*p* < 0.001). Also, 36.7% of the deceased group were male, compared to 27.7% in the alive group (*p* < 0.001). Race/ethnicity differed significantly, with 87.6% of the deceased group being non-Hispanic White, compared to 83.3% of the alive group (*p* = 0.035). A higher proportion of the deceased group (93.3%) was born in the U.S. compared to the alive group (88.8%; *p* = 0.002). Educational attainment varied significantly, with 58.7% of the deceased group having a high school degree or less compared to 46.7% in the alive group (*p* < 0.001). Marital status also showed significant differences: 48.6% of the deceased group were widowed, compared to 34.1% in the alive group (*p* < 0.001). Activity limitations were more common in the deceased group (81.4%) compared to the alive group (51.8%; *p* < 0.001). The distribution of chronic diseases was significantly different, with 35.8% of the deceased group reporting no chronic diseases compared to 32.2% in the alive group (*p* = 0.002). Repeat fall injuries were more common among the deceased group (7.0%) compared to the alive group (4.4%; *p* = 0.018). The deceased groups had significantly higher proportions of those who rated their health as poor (24.4%) compared to the alive group (8.5%; *p* < 0.001). Finally, activity intensity was significantly associated with mortality, as 89.3% of the deceased group had low activity intensity, compared to 73.9% in the alive group (*p* < 0.001).

In the unadjusted hazard model, respondents aged 75–84 years (HR: 2.30; 95% CI: 1.94–2.74) and 85 years and older (HR: 4.53; 95% CI: 3.77–5.44) had higher mortality risk, while females (HR: 0.74; 95% CI: 0.64–0.86) and Hispanics (HR: 0.73; 95% CI: 0.55–0.97) were associated with reduced mortality risk. Additionally, respondents with postgraduate education (HR: 1.40; 95% CI: 1.08–1.83), current smoking status (HR: 1.34; 95% CI: 1.04–1.72), functional limitations (HR: 3.03; 95% CI: 2.56–3.59), 3–4 chronic diseases (HR: 1.22; 95% CI: 1.03–1.45), five or more chronic diseases (HR: 1.67; 95% CI: 1.34–2.09), repeat falls (HR: 1.59; 95% CI: 1.25–2.03), poor (HR: 2.67; 95% CI: 2.24–3.19) and fair self-rated health (HR: 1.74; 95% CI: 1.49–2.03) had higher mortality risk (Table 3). Low activity intensity was associated with 99% increased mortality risk compared to normal-to-high activity intensity (HR: 1.99; 95% CI: 1.62–2.44). After adjusting for sociodemographic variables, health and injury status, and healthcare access factors, low activity intensity remained associated with a 50% increased mortality risk (Adjusted Hazard Ratio (aHR): 1.50; 95% CI: 1.20–1.87).

Consistently over time, the Kaplan–Meier survival curve for respondents with none-to-low activity intensity was significantly lower than that for those with normal-to-high activity intensity (Figure 3A). The median survival times were 8 years for the none-to-low group and 12 years for the normal-to-high group. Similarly, cumulative mortality hazard estimates were consistently higher among respondents with none-to-low activity intensity compared to those with normal-to-high activity intensity (Figure 3B). The mortality incidence rate was 8.5% in the none-to-low activity intensity group and 4.5% in the normal-to-high group. Respondents with none-to-low activity intensity had an 89% higher mortality incidence rate compared to those with normal-to-high activity intensity (IRR 1.89; 95% CI: 1.55–2.32). Additionally, 47% of the mortality in the none-to-low group could be attributed to low or no activity intensity.

For the sensitivity analysis, we re-categorized activity intensity into three levels (none, low, and normal-to-high) to assess whether separating participants with no activity from those with low activity altered the observed associations in the Cox regression models (Table 4). Compared with participants reporting normal-to-high activity, those with no activity had 46% increased mortality risk (aHR: 1.46; 95% CI: 1.15–1.83), while those with low activity had a 69% increased mortality risk (aHR: 1.69; 95% CI: 1.29–2.22). These findings indicate that both no and low activity intensity were independently associated with increased mortality risk, with no clear incremental difference between the two categories.

## 4. Discussion

In this study, we found that low activity intensity was significantly associated with an increased risk of mortality among older adults. The analysis revealed that individuals with low activity levels had nearly double the risk of death compared to those with normal-to-high activity intensity. Over the study period, the proportion of older adults reporting low activity levels decreased, suggesting a potential shift in physical activity patterns. In addition to activity levels, other important modifiable factors contributing to increased mortality risk included current smoking, activity limitations, and the presence of five or more chronic diseases. These findings highlight the crucial role of physical activity, in conjunction with other lifestyle factors, in the health and longevity of older adults.

Low activity levels may be associated with increased mortality following fall injury due to several interconnected factors. Older adults who engage in insufficient physical activity often experience reduced muscle strength, poor balance, and diminished coordination, all of which can heighten the risk of falling [19,41,42,43]. Additionally, sedentary behavior contributes to weakened bone density and increased frailty, making individuals more susceptible to serious injuries from falls, which can lead to higher mortality rates [44,45]. Physical activity, particularly exercises aimed at improving strength, balance, and flexibility, has been shown to decrease fall risk by enhancing these physical capabilities [6,19]. Furthermore, low activity levels can exacerbate other health conditions such as obesity, cardiovascular disease, and diabetes [46,47,48], which increase the occurrence of falls and fall-related injuries [49,50,51,52].

The decreasing proportion of older adults with low activity levels over time can be attributed to several factors, including public health interventions that promote exercise behavior [53,54,55], increased awareness of the importance of physical activity [56,57], and improved access to exercise programs tailored for older adults [58]. Over the past few decades, there has been a growing emphasis on promoting active lifestyles for aging populations through various campaigns, healthcare guidelines, and community-based programs aimed at mitigating the adverse effects of sedentary behavior [58]. As a result, older adults are more likely to engage in physical activities such as walking, strength training, and balance exercises, which have been shown to improve both mobility and overall health [58]. Additionally, healthcare providers are increasingly encouraging physical activity as a preventive measure for conditions such as cardiovascular disease, diabetes, and osteoporosis, leading to more older adults adopting healthier habits [59]. Accessibility and social support also play a critical role in the decreasing pattern of none-to-low activity intensity. Earlier studies have reported that the presence of parks, recreational facilities, and walk-friendly infrastructure, such as sidewalks and trails, contributes to a positive trend in activity levels [60,61].

While the decline in the proportion of older adults reporting none-to-low physical activity is a positive trend, the overall levels of activity remain unacceptably low. As of 2017, approximately 71% of older adults still reported none-to-low activity intensity, consistent with reports from the Centers for Disease Control and Prevention, which stated that fewer than 20% meet national aerobic activity guidelines [62]. To achieve meaningful progress, strategies must go beyond simply raising awareness of exercise benefits to encouraging behavioral change [57]. Interventions must be tailored and multifaceted—such as incorporating physical activity counseling into routine clinical care, expanding access to community-based exercise programs, leveraging technology (e.g., step counters, motivational text messaging), and addressing environmental and policy-level supports like walkable neighborhoods and affordable fitness facilities. SilverSneakers^®^ is an example of a resource for age-friendly exercise classes [63]. The program, covered by some Medicare plans, includes no-cost access to select gyms, as well as online on-demand and synchronous exercise classes. Special attention should be given to high-risk groups, including those with cardiovascular, cancer, diabetes, and dementia comorbid conditions. A coordinated approach that combines behavioral support, structural access, and clinical engagement is crucial for increasing physical activity and improving health outcomes in aging populations. For example, promising home-based intervention programs to increase physical activity among people with dementia include a variety of exercise components, home modifications, caregiver education and communication training, goal setting, and problem-solving [64].

This study has several limitations. First, this study is observational, and causality cannot be established. The observed relationship may reflect reverse causality, as individuals with poorer health may engage in lower levels of physical activity, which in turn contributes to higher mortality risks. There is a potential for nondifferential misclassification due to the reliance on self-reported physical activity and health status, which may be prone to bias [30,65]. Although we defined physical activity intensity using a standardized approach, there is still the possibility of participants underreporting or overreporting their activity levels. Mortality status, however, was obtained from the National Death Index, which is considered a reliable source and minimizes misclassification of the outcome. We were unable to control for all potential confounding factors, such as specific chronic conditions, injury severity, medication use, frailty index, cognitive function, depression, nutritional status, or genetic predispositions, which may influence mortality risk but were not included in the dataset. Also, we defined the chronic disease index using only the ten comorbid conditions available in the dataset, which prevented the use of standard mortality indices such as the Elixhauser or Charlson comorbidity index [66,67]. This restricted list of comorbid conditions likely explains the discrepancy between the low prevalence of chronic conditions and the high prevalence of activity limitations among those with none-to-low activity. Of note, many of the impairments used to define activity limitations include vision or hearing loss, injury, or developmental and mental health problems, and these measures are not captured as comorbid conditions [68]. Despite these limitations, the study provides valuable insights into the relationship between physical activity and mortality risk among older adults, highlighting the need for further research to explore these associations in greater depth using longitudinal data and more diverse populations.

## 5. Conclusions

This study highlights the association between none-to-low physical activity intensity and increased all-cause mortality following fall injuries among older adults. Although the proportion of older adults reporting none-to-low activity levels has declined over time, the majority remain insufficiently active, placing them at elevated mortality risk following fall injury. Our findings have several implications. At the individual level, there is a need for continued personalized exercise plans tailored to the specific needs of older adults, particularly those with chronic conditions or activity limitations. Community-level interventions should focus on further expanding access to affordable, age-friendly physical activity opportunities and reducing sedentary behavior through supportive social networks and infrastructure. Within healthcare settings, routine screening for physical activity levels and the integration of physical activity counseling into chronic disease management can play a preventive role. At the policy level, investments in public health campaigns and infrastructure improvements are essential to support active aging. Future research should explore pathways linking multi-level interventions to improved exercise behavior and associated health outcomes.

## Figures and Tables

**Figure 1 healthcare-13-02530-f001:**
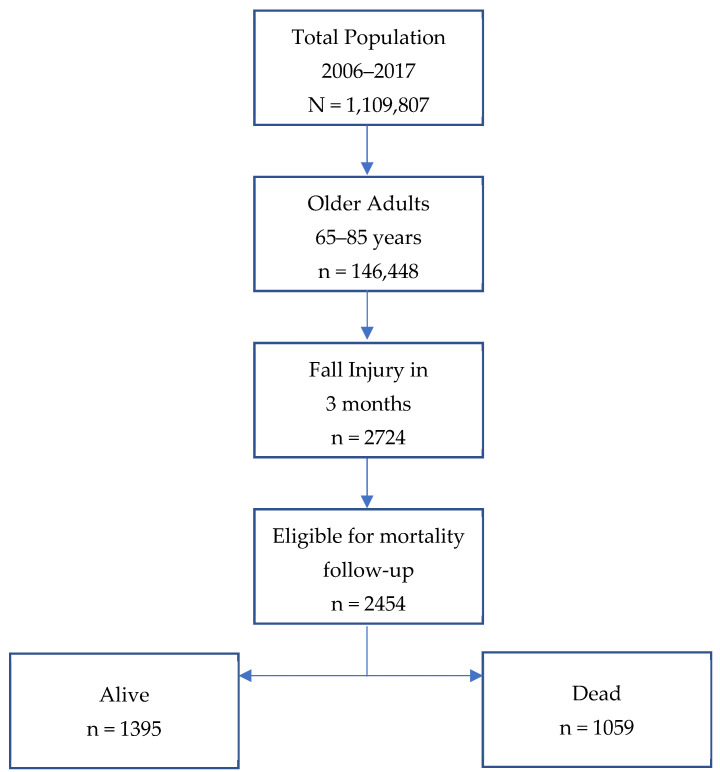
Data selection steps.

**Figure 2 healthcare-13-02530-f002:**
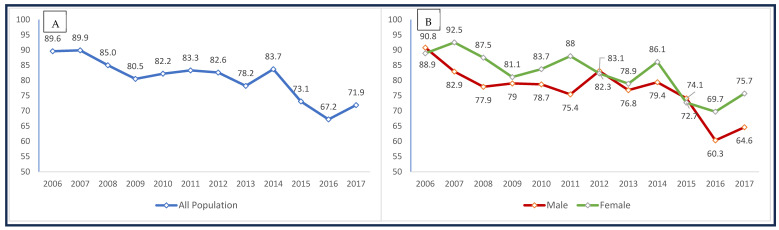
Trend in the proportion of none-to-low activity intensity between 2006 and 2017 among (**A**) all older adults and (**B**) male and female older adults.

**Figure 3 healthcare-13-02530-f003:**
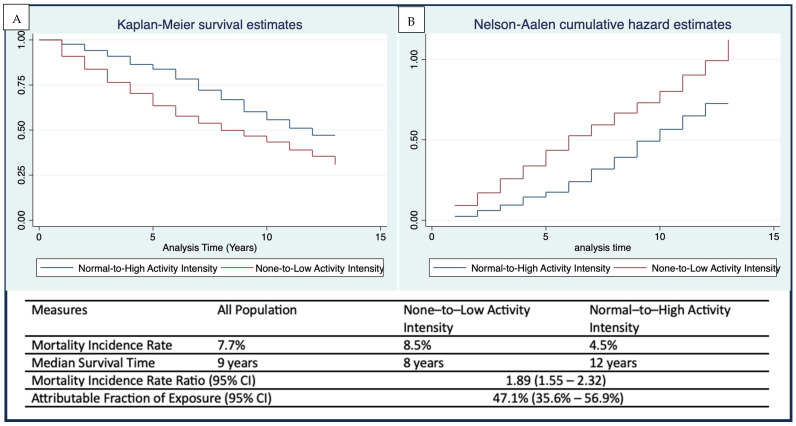
(**A**) Kaplan-Meier survival and (**B**) Nelson-Aalen cumulative hazard curves, including the mortality incidence, survival times, and mortality rate ratio among older adults with none-to-low and normal-to-high activity intensity.

**Table 1 healthcare-13-02530-t001:** Frequency distribution and association between activity intensity and sociodemographic, health, injury, health care access, and mortality status among older adults.

Variable	All Population Unweighted Count (Weighted %) N = 2454	None-to-Low Activity Unweighted Count (Weighted %) n = 1990 (80.8)	Normal-to-High Activity Unweighted Count (Weighted %) n = 464 (19.2)	*p*-Value
Mean Age (95% CI)	76.6 (76.3, 76.9)	77.0 (77.6, 77.4)	74.9 (74.2, 75.7)	
Age Category				
65–74 years	1026 (41.1)	772 (38.7)	254 (51.5)	<0.001
75–84 years	851 (34.7)	710 (35.2)	141 (32.3)	
85 years and older	577 (24.2)	508 (26.1)	69 (16.2)	
Sex				
Male	784 (31.6)	609 (30.2)	175 (37.2)	0.012
Female	1670 (68.4)	1381 (69.8)	289 (62.8)	
Race/Ethnicity				
Non-Hispanic White	1962 (85.2)	1581 (85.2)	381 (85.3)	0.929
Non-Hispanic Black	200 (6.4)	169 (6.5)	31 (5.9)	
Hispanic	173 (5.0)	145 (5.0)	28 (5.2)	
Other Races	119 (3.3)	95 (3.2)	24 (3.7)	
Born in the US				
Yes	2196 (90.8)	1781 (90.8)	415 (90.8)	0.998
No	258 (9.2)	209 (9.2)	49 (9.2)	
Educational Attainment				
High school or less	1302 (52.1)	1126 (55.8)	176 (36.4)	<0.001
Some college or AA	640 (26.0)	491 (24.4)	149 (32.6)	
Bachelor’s degree	296 (12.8)	219 (11.9)	77 (16.6)	
Postgraduate	216 (9.1)	154 (7.9)	62 (14.4)	
Marital Status				
Single	104 (4.5)	83 (4.1)	21 (6.0)	0.034
Married	994 (41.0)	807 (41.2)	187 (40.1)	
Divorced	359 (13.8)	273 (12.9)	86 (17.8)	
Widowed	997 (40.7)	827 (41.8)	170 (36.0)	
Poverty Level				
Poor	2186 (90.9)	1760 (90.8)	426 (91.4)	0.676
Not Poor	268 (9.1)	230 (9.2)	38 (8.6)	
Smoking Status				
Never Smoked	1613 (65.8)	1389 (69.9)	224 (48.4)	<0.001
Current Smoker	155 (6.2)	123 (6.0)	32 (6.9)	
Former Smoker	686 (28.0)	478 (24.1)	208 (44.7)	
No Available Care				
Yes	27 (1.1)	21 (1.0)	6 (1.3)	0.675
No	2427 (98.9)	1969 (99.0)	458 (98.7)	
No Accessible Care				
Yes	124 (5.1)	85 (4.4)	39 (7.8)	0.011
No	2330 (94.9)	1905 (95.6)	425 (92.2)	
No Affordable Care				
Yes	98 (3.9)	82 (4.1)	16 (3.0)	0.352
No	2356 (96.1)	1908 (95.6)	448 (97.0)	
Health Coverage				
Yes	6 (0.2)	5 (0.2)	1 (0.4)	0.393
No	2448 (99.8)	1985 (99.8)	463 (99.6)	
Activity Limitations				
Yes	1625 (65.2)	1420 (70.3)	205 (43.6)	<0.001
No	829 (34.8)	570 (29.7)	259 (56.4)	
Index of chronic disease				
None	820 (33.9)	781 (40.0)	39 (7.7)	<0.001
1–2	762 (31.1)	522 (25.6)	240 (54.4)	
3–4	663 (26.6)	512 (25.9)	151 (29.8)	
5 or more	209 (8.4)	175 (8.5)	34 (8.1)	
Repeat Fall Injury				
Yes	136 (5.6)	122 (6.2)	14 (3.0)	0.020
No	2318 (94.4)	1868 (93.8)	450 (97.0)	
Self-rated health				
Poor	391 (15.7)	375 (18.3)	16 (4.6)	<0.001
Fair	604 (24.3)	534 (26.6)	70 (14.8)	
Good to Excellent	1459 (60.0)	1081 (55.1)	378 (80.6)	
Mortality Status				
Dead	1059 (45.3)	941 (50.0)	118 (25.4)	<0.001
Alive	1395 (54.7)	1049 (50.0)	346 (74.6)	

**Table 2 healthcare-13-02530-t002:** Frequency distribution and association between mortality status and sociodemographic, health, injury, and health care access characteristics among older adults.

Variable	Dead Unweighted Count (Weighted %)	Alive Unweighted Count (Weighted %)	*p*-Value *
Mean Age (95% CI)	79.5 (79.0, 79.9)	74.2 (73.8, 74.6)	
Age Category			
65–74 years	259 (23.4)	767 (55.9)	<0.001
75–84 years	401 (37.5)	450 (32.3)	
85 years and older	399 (39.1)	178 (11.8)	
Sex			
Male	411 (36.7)	373 (27.7)	<0.001
Female	648 (63.3)	1022 (72.7)	
Race/Ethnicity			
Non-Hispanic White	871 (87.6)	1091 (83.3)	0.023
Non-Hispanic Black	80 (0.1)	120 (0.1)	
Hispanic	68 (0.0)	105 (0.1)	
Other Races	40 (0.0)	79 (0.0)	
Born in the US			
Yes	976 (93.3)	1220 (88.8)	<0.001
No	83 (6.7)	175 (11.2)	
Educational Attainment			
High school or less	622 (58.7)	680 (46.7)	0.011
Some college or AA	239 (22.5)	401 (28.8)	
Bachelor’s degree	120 (11.9)	176 (13.5)	
Postgraduate	78 (6.9)	138 (11.0)	
Marital Status			
Single	38 (4.1)	66 (4.8)	<0.001
Married	403 (37.6)	591 (43.8)	
Divorced	113 (9.7)	246 (17.3)	
Widowed	505 (48.6)	492 (34.1)	
Poverty Level			
Poor	107 (8.3)	161 (9.8)	0.179
Not Poor	952 (91.7)	1234 (90.2)	
Smoking Status			
Never Smoked	687 (66.1)	926 (65.5)	0.002
Current Smoker	79 (6.8)	76 (5.7)	
Former Smoker	293 (27.1)	393 (28.8)	
No Available Care			
Yes	13 (1.2)	14 (1.0)	0.477
No	1046 (98.8)	1381 (99.0)	
No Accessible Care			
Yes	51 (4.9)	73 (5.3)	0.581
No	1008 (95.1)	1322 (94.7)	
No Affordable Care			
Yes	39 (3.6)	59 (4.2)	0.333
No	1020 (96.4)	1336 (95.8)	
No Health Coverage			
Yes	1 (0.1)	5 (0.3)	0.127
No	1058 (99.9)	1390 (99.7)	
Activity Limitations			
Yes	868 (81.4)	757 (51.8)	<0.001
No	191 (18.6)	638 (48.2)	
Index of chronic disease			
None	376 (35.8)	444 (32.2)	<0.001
1–2	277 (27.1)	485 (34.5)	
3–4	292 (27.0)	371 (26.3)	
5 or more	114 (10.1)	95 (7.0)	
Repeat Fall Injury			
Yes	71 (7.0)	65 (4.4)	0.004
No	988 (93.0)	1330 (95.6)	
Self-rated health			
Poor	258 (24.4)	133 (8.5)	<0.001
Fair	304 (28.1)	300 (21.1)	
Good to Excellent	497 (47.5)	962 (70.4)	
Activity intensity			
None-to-Low	941 (89.3)	1049 (73.9)	<0.001
Normal-to-High	118 (10.7)	346 (26.1)	

* *p*-value determined using survey-weighted log-rank test.

**Table 3 healthcare-13-02530-t003:** Unadjusted and adjusted mortality risk ratios by metabolic activity, sociodemographic, health, injury, and healthcare access factors among older adults.

Variable	Unadjusted Hazard Risk Ratio (95% CI)	Adjusted Hazard Risk Ratio (95% CI)
Activity Intensity		
None-to-Low	**1.99 (1.62–2.44)**	**1.50 (1.20–1.87)**
Normal-to-High	Ref	Ref
Age Category		
65–74 years	Ref	Ref
75–84 years	**2.30 (1.94–2.74)**	**2.16 (1.81–2.58)**
85 years and older	**4.53 (3.77–5.44)**	**4.06 (3.33–4.95)**
Sex		
Male	Ref	Ref
Female	**0.74 (0.64–0.86)**	**0.64 (0.54–0.75)**
Race/Ethnicity		
Non-Hispanic White	Ref	Ref
Non-Hispanic Black	0.93 (0.72–1.19)	0.82 (0.66–1.09)
Hispanic	**0.73 (0.55–0.97)**	0.84 (0.70–1.16)
Other Races	**0.65 (0.43–0.97)**	**0.61 (0.40–0.94)**
Born in the US		
Yes	**1.76 (1.31–2.37)**	**1.76 (1.29–2.41)**
No	Ref	Ref
Educational Attainment		
High school or less	Ref	Ref
Some college or AA	1.20 (0.86–1.67)	**1.35 (1.00–1.83)**
Bachelor’s degree	1.10 (0.81–1.48)	1.08 (0.81–1.44)
Postgraduate	**1.40 (1.08–1.83)**	1.16 (0.90–1.51)
Marital Status		
Single	Ref	Ref
Married	0.84 (0.56–1.25)	0.86 (0.57–1.31)
Divorced	0.71 (0.46–1.09)	0.75 (0.48–1.18)
Widowed	1.34 (0.84–1.86)	0.88 (0.58–1.34)
Poverty Level		
Poor	0.91 (0.73–1.12)	0.82 (0.64–1.05)
Not Poor	Ref	Ref
Smoking Status		
Never Smoked	Ref	Ref
Current Smoker	**1.34 (1.04–1.72)**	**1.50 (1.13–1.99)**
Former Smoker	1.05 (0.90–1.23)	1.10 (0.92–1.32)
No Available Care		
Yes	1.09 (0.68–1.76)	0.71 (0.33–1.54)
No	Ref	Ref
No Accessible Care		
Yes	1.07 (0.78–1.46)	1.08 (0.78–1.50)
No	Ref	Ref
No Affordable Care		
Yes	0.84 (0.61–1.17)	0.82 (0.56–1.20)
No	Ref	Ref
No Health Coverage		
Yes	0.17 (0.02–1.34)	0.32 (0.04–2.79)
No	Ref	Ref
Activity Limitations		
Yes	**3.03 (2.56–3.59)**	**2.00 (1.66–2.41)**
No	Ref	Ref
Index of chronic disease		
None	Ref	Ref
1–2	0.97 (0.82–1.14)	1.06 (0.88–1.29)
3–4	**1.22 (1.03–1.45)**	1.05 (0.85–1.29)
5 or more	**1.67 (1.34–2.09)**	**1.47 (1.13–1.91)**
Repeat Fall Injury		
Yes	**1.59 (1.25–2.03)**	0.95 (0.71–1.27)
No	Ref	Ref
Self-rated health		
Poor	**2.67 (2.24–3.19)**	**2.07 (1.68–2.54)**
Fair	**1.74 (1.49–2.03)**	**1.41 (1.19–1.68)**
Good to Excellent	Ref	Ref

Significant results in bold fonts.

**Table 4 healthcare-13-02530-t004:** Sensitivity analysis showing the association between three-level activity intensity and mortality risk ratio in the unadjusted and adjusted survey-weighted Cox regression models.

Variable	Unweighted Count (Weighted %)	Unadjusted Hazard Risk Ratio (95% CI)	Adjusted Hazard Risk Ratio (95% CI)
Activity Intensity			
None	1732 (70.3)	**2.01 (1.63–2.47)**	**1.46 (1.15–1.83)**
Low	258 (10.6)	**1.85 (1.41–2.41)**	**1.69 (1.29–2.22)**
Normal-to-High	464 (19.1)	Ref	Ref

Significant results in bold.

## Data Availability

The original data presented in the study are openly available in FigShare at 10.6084/m9.figshare.29973412.

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
