# Peer review of "Activity Intensity and All-Cause Mortality Following Fall Injury Among Older Adults: Results from a 12-Year National Survey"

_healthcare, 2025, doi:10.3390/healthcare13192530_

Round 1
Reviewer 1 Report
Comments and Suggestions for Authors
The study investigates the relationship between activity intensity and fall-related deaths among community-dwelling U.S. older adults. The research team used data from the National Health Interview Survey and identified 2,454 older adults (65 years and older) who had sustained fall injuries.
The study controlled for sociodemographic, healthcare access, and health characteristics, and used survey-weighted Cox proportional hazard regression analysis to assess the relationship between activity intensity and fall-related death.
The results showed that approximately 81% of the study population had low activity levels. However, the proportion of the study population with low physical activity decreased from 90% to 67% between 2006 and 2017.
After adjusting for various characteristics, the study found that none-to-low activity intensity was associated with a 60% increased mortality risk (aHR: 1.60; 95% CI: 1.28 – 2.00).
The conclusion drawn from the study is that promoting higher physical activity levels may significantly reduce the risk of fall-related deaths among older adults. This highlights the importance of encouraging physical activity in this population to improve public health outcomes.
The study provides valuable insights into the relationship between activity intensity and fall-related deaths among older adults. However, it is important to note that the study is retrospective and relies on survey data, which may have limitations. Nevertheless, the findings underscore the need for interventions to promote physical activity among older adults to reduce the risk of falls and related deaths.
The inability to control for all potential confounding factors, such as specific chronic conditions, injury severity, medication use, or genetic predispositions, may influence mortality risk but were not included in the dataset.
The researchers combined 12 years of data from 2006 to 2017 using the Integrated Public Use Microdata Series (IPUMS) of the National Health Interview Survey (NHIS). IPUMS is a publicly accessible resource that harmonizes NHIS data, making it easier to combine data across years.
The NHIS is one of the oldest national surveys in the U.S., sampling over 35,000 households and 87,500 non-institutionalized individuals each year. Participants are randomly selected from all 50 states and the District of Columbia using a multistage sampling strategy. The annual household response rate is on average 70%, with oversampling of Black, Asian, and Hispanic populations to enhance the precision of the estimates.
Reliance on self-reported physical activity and health status introduces the potential for misclassification due to bias. Participants may underreport or overreport their activity levels, affecting the accuracy of the data.

Author Response
Comment
A brief summary (one short paragraph) outlining the aim of the paper, its main contributions and strengths.
The study investigates the relationship between activity intensity and fall-related deaths among community-dwelling U.S. older adults. The research team used data from the National Health Interview Survey and identified 2,454 older adults (65 years and older) who had sustained fall injuries. The study controlled for sociodemographic, healthcare access, and health characteristics, and used survey-weighted Cox proportional hazard regression analysis to assess the relationship between activity intensity and fall-related death. The results showed that approximately 81% of the study population had low activity levels. However, the proportion of the study population with low physical activity decreased from 90% to 67% between 2006 and 2017. After adjusting for various characteristics, the study found that none-to-low activity intensity was associated with a 60% increased mortality risk (aHR: 1.60; 95% CI: 1.28 – 2.00). The conclusion drawn from the study is that promoting higher physical activity levels may significantly reduce the risk of fall-related deaths among older adults. This highlights the importance of encouraging physical activity in this population to improve public health outcomes.
Response
Thank you. Your feedback has strengthened our manuscript. Based on the feedback of one of the other reviewers, our adjusted hazard risk is 1.50 and no longer 1.60.
Comment
The study provides valuable insights into the relationship between activity intensity and fall-related deaths among older adults. However, it is important to note that the study is retrospective and relies on survey data, which may have limitations. Nevertheless, the findings underscore the need for interventions to promote physical activity among older adults to reduce the risk of falls and related deaths. The inability to control for all potential confounding factors, such as specific chronic conditions, injury severity, medication use, or genetic predispositions, may influence mortality risk but were not included in the dataset.
Response
Thank you for your feedback. We added this concern in the limitations section. It reads:
“This study has several limitations. First, this study is observational, and causality cannot be established.”
Discussion: Paragraph 5
Comment
The researchers combined 12 years of data from 2006 to 2017 using the Integrated Public Use Microdata Series (IPUMS) of the National Health Interview Survey (NHIS). IPUMS is a publicly accessible resource that harmonizes NHIS data, making it easier to combine data across years.
The NHIS is one of the oldest national surveys in the U.S., sampling over 35,000 households and 87,500 non-institutionalized individuals each year. Participants are randomly selected from all 50 states and the District of Columbia using a multistage sampling strategy. The annual household response rate is on average 70%, with oversampling of Black, Asian, and Hispanic populations to enhance the precision of the estimates.
Response
Thank you for your comment.
Comment
Reliance on self-reported physical activity and health status introduces the potential for misclassification due to bias. Participants may underreport or overreport their activity levels, affecting the accuracy of the data.
Response
Thank you for noting this. We reported the same in our manuscript
“There is a potential for nondifferential misclassification due to the reliance on self-reported physical activity and health status, which may be prone to bias.[30,65] Although we defined physical activity intensity using a standardized approach, there is still the possibility of participants underreporting or overreporting their activity levels.”
Discussion: Paragraph 5
Comment
The article is written correctly, there are descriptions of the data, the language corresponds to the scientific style, and the authors refer to relevant literature.
Response
Thank you for your feedback
Reviewer 2 Report
Comments and Suggestions for Authors
The topic is clinically important and the dataset is large and nationally representative. The paper’s strengths include clear inclusion criteria, use of survey weighting, and survival analysis. However, issues need substantial clarification:
Major comments
The text refers to “fall-related death,” but analyses appear to use all-cause mortality. Please either (a) reframe title/abstract/results as all-cause mortality, or (b) re-analyze using cause-of-death codes (cause-specific Cox or Fine–Gray competing risks).
If available in NHIS, adjust for baseline self-rated health and functional difficulties (ADL/IADL items), and consider sensitivity analyses (e.g., excluding those with severe limitations at baseline; E-values to gauge unmeasured confounding)
If it is possible, avoid binary cut-offs (<500 vs ≥500 MET-min). Show dose–response (≥3–4 categories or splines) and check nonlinearity.
Scope/generalizability: Clarify that findings apply to older adults with recent injurious falls (title/abstract), and temper general claims.
Author Response
Comment
The topic is clinically important and the dataset is large and nationally representative. The paper’s strengths include clear inclusion criteria, use of survey weighting, and survival analysis. However, issues need substantial clarification:
Response
Thank you for your feedback. It has strengthened the manuscript.
Comment
The text refers to “fall-related death,” but analyses appear to use all-cause mortality. Please either (a) reframe title/abstract/results as all-cause mortality, or (b) re-analyze using cause-of-death codes (cause-specific Cox or Fine–Gray competing risks).
Response
The appropriate term is all-cause mortality following fall injury. We replaced “fall-related death” with “all-cause mortality following fall injury”. We made these changes in the title, abstract, and all relevant sections in the manuscript. In the Methods section, we clarified that death refers to all-cause mortality.
New Title: “Activity Intensity and All-Cause Mortality Following Fall Injury Among Older Adults: Results from A 12-Year National Survey”
“The outcome variable for this study was time to death among older adults. Death was defined as all-cause mortality occurring after a fall injury. The time component was determined as the difference between the year of mortality and the baseline year of observation. Participants who were alive at the end of the study period were right censored.”
Methods: Outcome Variable
Comment
If available in NHIS, adjust for baseline self-rated health and functional difficulties (ADL/IADL items), and consider sensitivity analyses (e.g., excluding those with severe limitations at baseline; E-values to gauge unmeasured confounding)
Response:
Thank you for your comment. The NHIS does not have ADL/IADL. However, it has questions about activity limitations. We controlled for activity limitations in the earlier version. We have added self-rated health to the model and updated the Methods and Results sections. The adjusted hazard risk was attenuated, and we edited all the numbers in all our tables.
“Health-related confounders included smoking status (never smoked, current smoker, and former smoker), activity limitations (binary variable – yes or no, defined as having any condition or health problems causing limitations in any way), chronic disease index (computed as sum of binary indicators (reported = 1, not reported = 0) of hypertension, diabetes, chronic obstructive pulmonary disease, asthma, angina, arthritis, cancer, coronary heart disease, liver disease, and peptic ulcer disease and recoded as none, 1–2, 3–4, and 5+ chronic conditions), history of repeat fall injury (yes or no), and self-rated health (defined as a three-level categorical variable – poor, fair, and good-to-excellent).”
Methods: Potential Confounders
After adjusting for sociodemographic variables, health and injury status, and healthcare access factors, low activity intensity remained associated with a 50% increased mortality risk (Adjusted Hazard Ratio (aHR): 1.50; 95% CI: 1.20–1.87).
Results: Paragraph 3
Comment
If it is possible, avoid binary cut-offs (<500 vs ≥500 MET-min). Show dose–response (≥3–4 categories or splines) and check nonlinearity.
Response
Thank you for this comment. We shared this concern as well. Hence, the reason we defined activity intensity as a binary and three-level categorical variable (see Table 3). In the earlier version, we reported in Table 4 the result of a three-level categorization. There was no difference in the 'none' and 'low' categories – while both are significant, the confidence intervals overlap. Our result, therefore, does not show a dose-response pattern, and we don’t intend to give that impression. Our choice of a binary classification was conservative because we cannot rule out recall bias that might have impacted the categorization, i.e., not all older adults will recall whether they engaged in “zero” physical activity or “low” physical activity. Here are the relevant sections:
“Individuals with <500 weekly MET-minutes were classified as none-to-low, while those with ≥500 MET-minutes were classified as normal-to-high. [28,29] We opted to define activity intensity as a binary variable to minimize variability due to recall bias.[30] However, we performed a sensitivity analysis reclassifying activity intensity as a three-level categorical variable – none, low, and normal-to-high (discussed below). For all the regression models, the "normal-to-high" category served as the reference group.”
Methods: Predictor Variable
Lastly, we performed a sensitivity analysis to observe any measurable differences in mortality risks among those classified as having no intense activity and low-intense activity. We reclassified activity intensity as a three-level categorical variable – none, low, and normal-to-high population and re-ran the survey-weighted adjusted hazard regression analysis.”
Methods: Data Analysis
Comment
Scope/generalizability: Clarify that findings apply to older adults with recent injurious falls (title/abstract), and temper general claims.
Response
We have tempered the claim that activity intensity is related to fall-related death. Instead, our manuscript now refers to all-cause mortality following a fall injury. For example, the conclusion in the abstract and the main manuscript read:
“Promoting higher physical activity levels may significantly reduce the all-cause mortality risk following fall injury among older adults.”
Abstract: Conclusion
“This study underscores the association between none-to-low physical activity intensity and increased all-cause mortality risk following fall injury among older adults.”
Conclusion
Reviewer 3 Report
Comments and Suggestions for Authors
1. Throughout the manuscript, the results are presented with a causal implication of “low activity → increased mortality.” However, reverse causality (e.g., individuals with poorer health engage in less physical activity, thereby increasing mortality risk) cannot be ruled out. Please clarify this point explicitly.
2. The adjustment variables are limited to general indicators such as age, sex, education, and number of chronic diseases. However, several important factors that may influence falls and mortality are missing (e.g., medication use, frailty index, fall severity, cognitive function, depression, nutritional status).
3. Throughout the manuscript, it is unclear whether the outcome represents “fall-related mortality” or “all-cause mortality after fall injury.” Please clarify this.
Author Response
Comments and Suggestions for Authors
- Throughout the manuscript, the results are presented with a causal implication of “low activity → increased mortality.” However, reverse causality (e.g., individuals with poorer health engage in less physical activity, thereby increasing mortality risk) cannot be ruled out. Please clarify this point explicitly.
Response
Thank you for your comment. We have addressed the issue of perceived causality in the limitations section. The added section reads:
“This study has several limitations. First, this study is observational, and causality cannot be established. The observed relationship may reflect reverse causality, as individuals with poorer health may engage in lower levels of physical activity, contributing to higher mortality risk.”
Discussion: Paragraph 5
Comment
- The adjustment variables are limited to general indicators such as age, sex, education, and number of chronic diseases. However, several important factors that may influence falls and mortality are missing (e.g., medication use, frailty index, fall severity, cognitive function, depression, nutritional status).
Response
We acknowledged this limitation in the original version, but we have updated the list to include these additional factors. The edited sentence reads:
“We were unable to control for all potential confounding factors such as specific chronic conditions, injury severity, medication use, frailty index, cognitive function, depression, nutritional status, or genetic predispositions, which may influence mortality risk but were not included in the dataset.”
Discussion: Paragraph 5
Comment
- Throughout the manuscript, it is unclear whether the outcome represents “fall-related mortality” or “all-cause mortality after fall injury.” Please clarify this.
Response
We have made the changes. The appropriate term is all-cause mortality following fall injury. We replaced “fall-related death” with “all-cause mortality following fall injury”. We made these changes in the title, abstract, and all relevant sections in the manuscript. In the Methods section, we clarified that death refers to all-cause mortality.
New Title: “Activity Intensity and All-Cause Mortality Following Fall Injury Among Older Adults: Results from A 12-Year National Survey”
“The outcome variable for this study was time to death among older adults. Death was defined as all-cause mortality occurring after a fall injury. The time component was determined as the difference between the year of mortality and the baseline year of observation. Participants who were alive at the end of the study period were right censored.”
Methods: Outcome Variable
Round 2
Reviewer 2 Report
Comments and Suggestions for Authors
The authors have adequately addressed all substantive concerns raised during peer review.